# Australian Public Perspectives on Genomic Newborn Screening: Risks, Benefits, and Preferences for Implementation

**DOI:** 10.3390/ijns10010006

**Published:** 2024-01-17

**Authors:** Fiona Lynch, Stephanie Best, Clara Gaff, Lilian Downie, Alison D. Archibald, Christopher Gyngell, Ilias Goranitis, Riccarda Peters, Julian Savulescu, Sebastian Lunke, Zornitza Stark, Danya F. Vears

**Affiliations:** 1Biomedical Ethics Research Group, Murdoch Children’s Research Institute, Parkville, VIC 3052, Australia; fiona.lynch@mcri.edu.au (F.L.); christopher.gyngell@mcri.edu.au (C.G.); jsavules@nus.edu.sg (J.S.); 2Melbourne Law School, The University of Melbourne, Melbourne, VIC 3052, Australia; 3Sir Peter MacCallum Cancer Centre Department of Oncology, University of Melbourne, Melbourne, VIC 3052, Australia; stephanie.best@petermac.org; 4Australian Genomics, Melbourne, VIC 3052, Australia; ilias.goranitis@unimelb.edu.au (I.G.); zornitza.stark@vcgs.org.au (Z.S.); 5Department of Health Services Research, Peter MacCallum Cancer Centre, Melbourne, VIC 3052, Australia; 6Murdoch Children’s Research Institute, Parkville, VIC 3052, Australia; clara.gaff@melbournegenomics.org.au (C.G.); lilian.downie@vcgs.org.au (L.D.); alison.archibald@vcgs.org.au (A.D.A.); 7Melbourne Genomics, Melbourne, VIC 3052, Australia; 8Department of Paediatrics, Faculty of Medicine, Dentistry and Health Sciences, The University of Melbourne, Melbourne, VIC 3052, Australia; 9Victorian Clinical Genetics Services, Murdoch Children’s Research Institute, Parkville, VIC 3052, Australia; sebastian.lunke@vcgs.org.au; 10Melbourne School of Population and Global Health, The University of Melbourne, Melbourne, VIC 3052, Australia; riccarda.peters@unimelb.edu.au; 11Centre for Biomedical Ethics, Yong Loo Lin School of Medicine, National University of Singapore, Singapore 117597, Singapore; 12Department of Pathology, The University of Melbourne, Melbourne, VIC 3052, Australia; 13Centre for Biomedical Ethics and Law, Department of Public Health and Primary Care, KU Leuven, 3000 Leuven, Belgium

**Keywords:** newborn screening, bioethics, genomic sequencing, qualitative, public views

## Abstract

Recent dramatic reductions in the timeframe in which genomic sequencing can deliver results means its application in time-sensitive screening programs such as newborn screening (NBS) is becoming a reality. As genomic NBS (gNBS) programs are developed around the world, there is an increasing need to address the ethical and social issues that such initiatives raise. This study therefore aimed to explore the Australian public’s perspectives and values regarding key gNBS characteristics and preferences for service delivery. We recruited English-speaking members of the Australian public over 18 years of age via social media; 75 people aged 23–72 participated in 1 of 15 focus groups. Participants were generally supportive of introducing genomic sequencing into newborn screening, with several stating that the adoption of such revolutionary and beneficial technology was a moral obligation. Participants consistently highlighted receiving an early diagnosis as the leading benefit, which was frequently linked to the potential for early treatment and intervention, or access to other forms of assistance, such as peer support. Informing parents about the test during pregnancy was considered important. This study provides insights into the Australian public’s views and preferences to inform the delivery of a gNBS program in the Australian context.

## 1. Introduction

Newborn bloodspot screening (NBS) is a highly successful population screening initiative in many parts of the world [1]. Developed by Robert Guthrie in the 1960s, the first NBS test used a blood sample collected from a heel prick onto filter paper (now known as the Guthrie card) to test for phenylketonuria [2]. While some conditions (such as congenital hypothyroidism and cystic fibrosis) were added in the 1970s and 80s [3], the 1990s saw an expansion of the number of conditions included in NBS panels with the introduction of tandem mass spectrometry technology [4]. Today, NBS programs around the world are highly successful, operating in many countries at relatively low cost with near-universal uptake [5]. The level and detail of informed consent required for traditional NBS varies by jurisdiction; in many countries, consent for traditional NBS is implied [6], yet overall, population trust and uptake remain high [7].

Standard NBS (stdNBS) acts as a screening test, meaning subsequent diagnostic testing is usually required to confirm a diagnosis [7]. In some cases, the diagnostic testing is genetic in nature; such NBS programs typically follow a protocol of first-tier biochemical testing (commonly tandem mass spectrometry), followed by genetic or genomic sequencing as a second-tier confirmatory/diagnostic test as appropriate, such as for cystic fibrosis [5,8,9]. However, some countries are either considering or beginning to include genetic testing as part of the first-line testing in NBS for conditions, such as spinal muscular atrophy [10,11].

Recent dramatic reductions in the timeframes in which genomic sequencing can deliver results, as well as the price, mean its application in time-sensitive screening programs such as NBS is becoming a reality [7]. Genomic newborn screening (gNBS) provides a single assay for the testing of many genetic conditions that would benefit from detection in infancy, but which do not have biochemical markers measurable by traditional NBS methodologies [12].

gNBS has sparked considerable discussion in the literature. Expert ethicists and clinicians in the field have suggested that we may not yet be ready for the implementation of gNBS [7,13] and position statements from leading bodies worldwide also urge caution at this time [14,15,16,17,18]. Yet, inclusion of genomic sequencing in NBS programs is not only inevitable [19,20], but already occurring [21]. Jurisdictions including the US (through the Guardian study, BabySeq, BeginNGS, and Early Check), England (Generation Study), Belgium (Baby Detect), France (PERIGENOMED), the EU (Screen4Care), and Australia (BabyScreen+) are all piloting gNBS programs as part of research studies [22,23].

To date, the inclusion of conditions in NBS programs has been based on the goal of identifying newborns with serious conditions who are likely to benefit from treatment early in life [22,24]. Yet with gNBS, the potential to screen for more conditions means that these parameters are being questioned [25,26,27].

Some work has been conducted using a variety of methodologies exploring parents’ and the public’s preferences and perspectives on gNBS in the US [20,28,29,30,31,32,33,34,35,36,37,38,39,40,41], Canada [42,43,44], the UK [45,46] and New Zealand [47]. Surveys have been conducted with Australian parents exploring their views [48,49]; however, to date, no in-depth qualitative work has been conducted with members of the Australian public.

As gNBS programs are developed around the world, there is an increasing need to address the ethical and social issues that such initiatives raise [22]. This study therefore aimed to explore the Australian public’s perspectives and values regarding key gNBS characteristics and preferences for gNBS service delivery.

## 2. Materials and Methods

We recruited English-speaking members of the Australian public over 18 years of age. Advertisements were distributed via social media posts on Facebook. Individuals who were interested in participating were directed to a website where they could register their interest and provide contact details. FL then telephoned potential participants to ask a series of screening questions to confirm eligibility and maximise diversity in focus group participation (e.g., diversity of age, gender, parental status, and geographical distribution). Screened individuals were sent further information about the study via email, invited to sign an online consent form, and provide their availabilities if they wished to participate. Participants were provided with a AUD 75 voucher as remuneration for their time at focus group completion.

Sampling aimed for heterogeneity in participant characteristics, including age; gender; location (by state and metropolitan/rural location); parental status; and country of birth and language spoken at home (as measures of cultural and linguistic diversity). Recruitment continued until sufficient heterogeneity was achieved and until minimal new data were generated addressing the study aims.

Prior to the focus groups, participants were asked to watch a 3 min video [50] to provide them with background information about the topics that would be discussed. The video included information about what DNA and genomic sequencing are, and how genomic sequencing can lead to disease detection. It also outlined what stdNBS in Australia involves and briefly introduced the idea of using genomics in NBS. Focus groups explored participants’ preferences and values regarding key gNBS characteristics and preferences for gNBS service delivery. The degree to which concepts such as the uncertainty and sensitivity of gNBS, targeted versus broader testing approaches, and the potential test outcomes were covered varied between focus groups depending on the discussion. The focus group guide is included as Appendix A.

Focus groups were facilitated by DV with support from FL and conducted via Zoom. Both DV and FL are skilled qualitative researchers with experience in focus group methodology and training in genetic counselling. None of the participants were known to the researchers.

Focus groups were recorded using the Zoom recording function and transcribed by FL. Interview transcripts were analysed by using inductive context analysis, whereby content categories are generated from the data, rather than predetermined [51]. Coding continued iteratively until all data relevant to the research question had been coded into categories and subcategories. Findings were discussed by DV and FL to ensure rigour. Data analysis was managed using NVivo (released March 2023) [52].

This study was reviewed and approved by The Royal Children’s Hospital Human Research Ethics Committee (HREC ID: 91392). Participants provided voluntary, informed consent.

## 3. Results

### 3.1. Participant Demographics

We had 643 people express interest in the study and managed to contact and confirm eligibility in 155 of them. No potential participants were excluded.

Seventy-five members of the Australian public aged 23–72 participated in 1 of 15 focus groups (range 2–8 per group). The participant characteristics are summarised in Table 1. We had representation from 7/8 states/territories across Australia. Thirteen participants self-reported they were medical or allied health professionals. Seventeen participants disclosed they had a child with a genetic condition. Eleven participants were on parental leave at the time of the focus group.

### 3.2. General Support for gNBS

Below, we present quantitative and qualitative data. Participants were polled at the beginning and end of each focus group to both facilitate discussion, and gauge if perspectives on gNBS changed over the course of the discussion. Quotes are used to illustrate our qualitative findings. An ellipsis (…) reflects where a significant portion of speech has been removed, and square brackets represent where a word has been replaced for clarity or to protect participant anonymity. Quotes are deidentified to protect participant anonymity; codes are used to identify participants based on their focus group number (e.g., FG1 P1 refers to focus group 1, participant 1).

All focus groups were asked whether they thought genomics should be used in newborn screening programs. Prior to the discussion, the majority (77%) of participants thought that genomics should be used, with 23% stating they were unsure. Following the focus group, the same number of participants thought that genomics should be used, however, in contrast to the beginning of the session, a proportion (7%) thought it should not be used.

Half of the focus groups were asked whether a NBS program that includes genomics should be run any differently to stNBS programs. Just under half (46%) of the participants expressed the view that a gNBS program should be run differently to a stdNBS program prior to the discussion. After the discussion, most (74%) thought it should be run differently.

The results of the polls are shown in Table 2.

Qualitative data showed that participants were generally supportive of introducing genomic sequencing into newborn screening, with several stating that the adoption of such revolutionary and beneficial technology was a moral obligation.


*“Personally, I think if the technology is there for us to screen and find anomalies of whatever sort early, I think we should use it.” [FG3 P1]*


### 3.3. Perceived Benefits of gNBS

Participants consistently highlighted receiving an early diagnosis as the leading benefit of implementing gNBS (Table 3, Quote 1). This was frequently linked to the potential for early treatment and intervention, or to access other forms of assistance, such as peer support (Table 3, Quote 2).

Participants also reflected on either their own or friends/families’ experiences of the diagnostic odyssey, suggesting that receiving an earlier diagnosis would reduce the stress associated with lengthy diagnoses (Table 3, Quote 3).

Furthermore, participants foresaw that receiving an early diagnosis would allow parents more time to prepare, both emotionally and practically (Table 3, Quotes 4 and 5).

Participants highlighted the potential benefits of gNBS to research and the ability to increase knowledge about rare genetic conditions (Table 3, Quote 6). They mentioned that an improved understanding of genetic conditions would lead to better resource allocation, such as government funding, to support patients and families with these conditions (Table 3, Quote 7).

Benefits of gNBS for other family members were also raised by participants. This included the ability to inform parents, siblings, and extended family members that there was a genetic condition within the family and explain how this knowledge could be used in pregnancy planning (Table 3, Quotes 8 and 9).

### 3.4. Potential Challenges with gNBS

Participants were concerned about the potential psychological impact that a diagnosis from gNBS could have on parents of newborns. They reflected that having a newborn was already a stressful time for new parents and were concerned about the added stress that an early genetic diagnosis might have on the family (Table 4, Quote 1). This was particularly the case for uncertain gNBS results and untreatable conditions (Table 4, Quote 2).

In addition to psychological impact, participants recognised that a diagnosis in a child might also have clinical implications for the parents as asymptomatic or mildly affected carriers of the condition (Table 4, Quote 3).

Relatedly, some participants discussed the potential negative impact of a diagnosis from gNBS on parent–child bonding (Table 4, Quote 4).

Participants expressed concern for the security of their and their child’s genomic data following gNBS (Table 4, Quote 5). They were particularly concerned about how the data might be used to discriminate against individuals in terms of insurance and employment later in life (Table 4, Quote 6). Because of this, they expressed that there should be strict regulations and processes for how genomic data are stored, shared, and accessed.

### 3.5. When to Discuss and Obtain Consent for gNBS

Most participants agreed that the availability of gNBS should be raised and discussed during pregnancy, rather than at, or soon after, the birth of the child (Table 5, Quote 1). Participants noted that birth and the immediate newborn period was a busy and stressful time for parents, and that this was therefore not an ideal time to be discussing something as complex at gNBS.

The exact timepoint at which gNBS should be discussed with soon-to-be parents raised many differing opinions. Some participants thought that gNBS should be discussed as early as possible, with some even suggesting implementing public education campaigns or raising it with couples trying to conceive (Table 5, Quotes 2 and 3).

However, others commented that various stages of pregnancy—from the first confirmation of pregnancy appointment to sometime in the third trimester—were appropriate times to have the conversation (Table 5, Quotes 4 and 5).

Some suggested that gNBS should be raised and discussed multiple times and potentially by multiple specialists to allow parents to fully understand the screening process (Table 5, Quote 6).

For similar reasons, many participants articulated that formal consent should also be obtained prior to birth (Table 5, Quote 7).

Some participants raised the concept of implementing two stages of consent, whereby parents provided formal, written consent prior to the birth, but that there was a process of ‘rechecking’ consent at the time of sample collection (Table 5, Quote 8).

### 3.6. Who Should Discuss gNBS

Participants were asked who they thought gNBS should be discussed by. Many suggested members of the existing prenatal care team, including the general practitioner (GP), obstetrician, midwife, hospital staff, or maternal and child health nurse (Table 5, Quotes 9, 10 and 11).

It was expressed that familiarity and rapport were important to have with the person discussing gNBS, and so someone that parents already have an existing relationship with would be best placed to discuss gNBS.

Some participants raised concerns that GPs were not adequately trained to discuss genomics or rare diseases with their patients.

Despite commonly proposing obstetricians or midwives as the most appropriate health professional to discuss gNBS, some participants were concerned that midwives might not have the training required to discuss gNBS in detail.

Other participants commented that discussions about gNBS required specialist expertise from someone like a genetic counsellor (Table 5, Quote 12).

Irrespective of who was responsible for discussing gNBS with parents, participants recognised that they needed to have the skills and training to provide accurate and comprehensive information, and to answer any questions that might arise (Table 5, Quote 13).

### 3.7. What Parents Need to Know about gNBS

Participants generally expressed that as much information as possible should be given to parents, and that discussions about gNBS should be detailed (Table 5, Quote 14).

However, others recognised that parents might be overwhelmed by receiving too much information, and instead, the detail of informed consent discussions should be tailored to the needs of each individual parent and family (Table 5, Quote 15).

There were also specific aspects of the screening test that participants thought parents should be made aware of. These included things such as which conditions were being tested for and which were not (Table 5, Quote 16).

With regard to processes, participants explained that parents needed to be made aware of what happens after gNBS, either in the case of a negative or positive result. This included an explanation of the timeframe, including when to expect to hear from a health professional regarding the child’s result (Table 5, Quote 17).

Participants agreed that parents should know what would happen to their child’s data, including where it would be stored, for how long, and who would have access to it (Table 5, Quote 18).

Participants wanted other resources to support parents in understanding and making a decision about gNBS for their child. As well as sufficient opportunities to ask questions of their health professional, participants suggested pamphlets, websites, a hotline, or even a genetic counsellor as alternative sources of information about gNBS (Table 5, Quote 19).

### 3.8. Type of Consent

When asked what type of consent gNBS should require, participants compared this with their experiences of consent for other tests and investigations (Table 6, Quote 1).

Several suggested that the choice about whether to have gNBS should be like the choice about whether to vaccinate against COVID-19. Others recognised that many things happen in hospital without explicit parental consent, and questioned whether gNBS should be any different. Yet some suggested that because most things for children did in fact require explicit consent, so should gNBS.

Reflecting specifically on their experience of stdNBS, most parents believed that consent for gNBS should be different (Table 6, Quote 2). Those that believed it should be the same assumed that existing consent processes for stdNBS were robust and in-depth (Table 6, Quote 3).

Some groups discussed whether gNBS should be mandated, whereby parents would have no choice but for their child to receive screening (Table 6, Quote 4). However, most thought that this was not appropriate, and that parents should instead be given the opportunity to opt-out (Table 6, Quote 5).

Irrespective of a parent’s decision, participants thought that they should be able to withdraw their consent at any time in the process (Table 6, Quote 6).

Participants generally agreed that gNBS should require explicit and informed consent from parents prior to being administered (Table 6, Quote 7). They explained that choice was important in both protecting parents’ autonomy, but also in facilitating trust in a population-based screening program.

Whether both parents needed to consent for gNBS for it to occur was a contentious issue in the focus group discussions. Some participants demonstrated strong views that both parents should be involved (Table 6, Quote 8). However, others ecognized that this was not always practical or possible, and that therefore only one parent’s consent should be required (Table 6, Quote 9).

Illustrative quotes are shown in Table 6.

## 4. Discussion

Our findings show that, after education and discussion, the members of the Australian public we engaged with were generally supportive of including genomic sequencing in NBS. This is in line with other studies reporting positive public and parental attitudes towards, and high interest in, gNBS, such as in the US, Canada and New Zealand [22,53]. Likewise, Genomics England demonstrated wide support for implementing genomic sequencing into NBS programs in their large-scale public deliberative approach [45,54]. Surveys with parents of healthy children enrolled in a randomised clinical trial of gNBS in the US also showed general support for population-wide implementation of gNBS for every newborn [39], and several studies report that parents express hypothetical interest in receiving genomic information about their newborn [29,30]. Of note, our participants thought a gNBS program should be run differently to stdNBS, which mainly related to the consent process as we discuss further below.

### 4.1. There Is Support for gNBS Provided Potential Risks Can Be Addressed

Participants in our study saw many benefits of gNBS, including early diagnosis, progression of medical research, and wider family benefits. However, participants were also cognisant of several potential risks associated with gNBS, such as data security, privacy, and the potential for discrimination. Research with parents and clinicians in the US demonstrate that both groups perceive both benefits and risks associated with gNBS; one of the major risks identified is the potential for psychological distress as a result of screening [20,37]. Our participants were also worried about the potential negative psychological impact of an early genetic diagnosis on parents and on parent–child bonding. Reassuringly, recent research from the BabySeq Project shows that families experience no sustained negative psychosocial effects from gNBS [55,56]. Interestingly, previous work suggests that parents perceive greater benefits and fewer risks associated with gNBS than clinicians [20,37], implying that our concerns may not be reflected by the wider public.

The concerns for genomic data security and privacy, and potential avenues for discrimination in employment and insurance raised by our participants are similar to those identified in a survey with parents conducted in the US [20]. Because of the very nature of gNBS programs, it is difficult to predict what changes in regulation may occur throughout the lifetime of today’s newborns, raising challenges for ensuring informed consent and maintaining trust in such programs. Relatedly, others have raised concerns about the potential infringement of the child’s autonomy by removing the child’s ability to make their own decisions about knowing their genomic information later in life [37].

### 4.2. Explicit Consent Should Be Required for gNBS

Although others in the literature have raised the timing of information provision and consent for gNBS as a concern [22], to our knowledge, this is the first study to gauge public opinion on the issue. Focus group participants agreed overwhelmingly that gNBS should be raised and discussed during pregnancy, probably more than once by multiple practitioners, and that at least a first stage of consent should also be implemented before birth.

Participants explained that a familiar face, that is, a member of the existing care team, would be best placed to have these discussions about gNBS with prospective parents. However, they raised concerns for the level of knowledge and skill required to have these conversations, suggesting that some professional education or training is required prior to the implementation of gNBS. Other population screening programs—such as expanded reproductive carrier screening—raise similar issues with health professional preparedness and a need for training [57]. In fact, the widespread adoption of genomic technologies across many specialty areas of medicine is generating considerable discussion about the need to upskill non-genetics health professionals to support patients [58]. Other authors have suggested that midwives in particular will play a significant role in the delivery of gNBS, and that any changes to NBS practice should therefore include multifaceted education programs aimed at such key stakeholders [59].

Participants in our study desired tailored informed consent discussions, covering as much or as little detail about gNBS as each parent required, which corresponds with a tiered and layered approach suggested by others in relation to diagnostic genomics [60]. There were, however, some key aspects that participants expressed should be communicated to all parents, including which conditions were being screened; the process following gNBS (particularly if a result suggesting a high chance for a genetic condition was identified); and what would happen to the child’s data. Participants also expressed that additional resources in varied formats could be useful to communicate complex information to families.

In Australia, NBS is voluntary and free for all infants, with uptake being close to 100% [49]. For gNBS, parents are likely to require more information to ensure that consent is informed and to maintain such high trust and coverage [7]. One concern is that implementing a more detailed and potentially burdensome informed consent process into NBS would reduce participation, thereby reducing the intended benefits of the program overall [36]. Although our participants appreciated this, they maintained that explicit, informed consent should be required for gNBS (rather than, for example, implicit consent or mandated screening) to facilitate trust in the program. Surveys with parents enrolled in a randomised clinical trial of gNBS in the US also suggest that parents see informed consent as more important for gNBS than for stdNBS [39]. Reassuringly, early studies obtaining explicit informed consent do not report a reduction in the uptake of gNBS compared to stdNBS [36,48,53]. However, some have suggested that introducing gNBS initially as a separate screening test to stdNBS may mitigate some of this risk [22].

### 4.3. Members of the Public Hold Less Conservative Views than Health Professionals

Our data support the general trend that views of the public about gNBS are less conservative than those of health professionals, who frequently report that gNBS is not currently ready for population-wide implementation [13,49,61,62,63]. Previous qualitative work also shows that while Australian healthcare professionals do not feel it is currently appropriate to incorporate genomic sequencing into NBS, they believe it will be implemented in the next decade [13].

One possible reason for this discrepancy is that parents and the public do not fully appreciate the complexities and potential challenges relating to gNBS. Certainly, one potential limitation of this study is that we deliberately provided limited information about how gNBS might be implemented prior to the focus groups so as not to bias their views. Topics such as the possibility of identifying variants of uncertain significance or late onset forms of disease using gNBS could be challenging for members of the public to understand. However, our participants raised many of the issues that concern HPs, indicating they do appreciate the nuances to some degree. This may suggest that perhaps HPs are overly concerned about these issues, or indicate higher levels of genetic exceptionalism within healthcare workers than the public. However, implementation of any gNBS program should consider the desires of the general population alongside those of experts.

## 5. Conclusions

Genomic sequencing has the potential to drastically increase the scope of existing NBS programs. Including genomic sequencing in NBS programs could help to identify conditions that traditional biochemical screening cannot [7] and overcome some of the current limitations of biochemical screening in premature and unwell infants [22]. Furthermore, the population-wide genomic data that would be generated from gNBS would also allow for research into the diagnosis, treatment and management of genetic diseases, as well as allow early access to clinical trials for those diagnosed through gNBS [22].

Implementation of such an ambitious population-wide screening program as gNBS is being considered in several contexts around the world. However, doing so without addressing parental concerns is likely to reduce trust and subsequently uptake of the program [39]. Engaging stakeholders in potential changes to NBS programs is therefore vital in maintaining a high participation rate [49] as well as in informing the design of programs so that they are tailored to local contexts.

## Figures and Tables

**Table 1 IJNS-10-00006-t001:** Participant characteristics.

	*n* (%)
Age	
Mean	42
Range	23–72
Metro/rural	
Metro	61 (81%)
Regional/rural	14 (19%)
Children	
Y	62 (83%)
N	13 (17%)
Country of birth	
Australia	61 (81%)
Other	14 (19%)
Language spoken at home	
English only	60 (80%)
Language other than English	15 (20%)
Gender	
Female	66 (88%)
Male	9 (12%)
TOTAL	75

**Table 2 IJNS-10-00006-t002:** Results of the poll.

		Pre-Discussion *n* (%)	Post-Discussion *n* (%)
Do you think genomics should be used in newborn screening programs?	Yes	58 (77%)	57 (77%)
	No	0	5 (7%)
	Not sure	17 (23%)	12 (16%)
	TOTAL	75	74
Should a newborn screening program that includes genomics be run any differently to standard newborn screening programs?	Yes	18 (46%)	29 (74%)
	No	9 (23%)	8 (21%)
	Not sure	12 (31%)	2 (5%)
	TOTAL	39	39

**Table 3 IJNS-10-00006-t003:** Benefits of gNBS.

Perceived Benefit	Illustrative Quote(s)
Benefits of early diagnosis	*Quote 1: “I think the other thing I’m really conscious of is that there will be medical and pharmaceutical interventions and therapeutic interventions in relation to some genetic conditions which, if undertaken at the earliest opportunity, will have more positive implications for children and their families in the long term.” [FG1 P8]* *Quote 2: “And also, getting support from other families…A lot of groups are on Facebook and things like that these days…you can hear how other people have gone through it and how they’ve survived it and all those sorts of things, or how they’re coping with it and what they’re using to cope with it and those sorts of things.” [FG9 P3]* *Quote 3: “…you can go through years of testing and hospitalisation trying to find an answer to a question about what’s going on with your child, and that would certainly make it easier if that could be available early on.” [FG3 P1]* *Quote 4: “I also imagine that for families that have to care for a loved one in this space, if they know ahead of time that there’s going to be a financial burden on them, it allows them to plan a little bit better if they know earlier.” [FG4 P6]* *Quote 5: “I kind of think knowledge is power regardless of what it is. Like you might not be able to alter things medically or like get the right treatment, whether or not it’s like the right kind of changes to make for your family, whether you’re a working mum and you decide to stop working or you’ve got other children that you need to care for and you might need some extra care. Might not necessarily be able to like make a decision medically about how to help your child, but how to help your own lifestyle and your own family. I think there’s other kinds of decisions that you can make around like news.” [FG15 P3]*
Research and knowledge benefits	*Quote 6:“I think it’s important because it gives an indication of actually how prevalent it is and that can help drive the funding and the research into it. If it’s not necessarily monitored then we don’t know just how rare it is…” [FG8 P5]* *Quote 7: “I’d like to think that the data could be used for more research but also for governments to plan and services to plan and if there’s going to show that there’s going to be more of particular disorders or illnesses, then that can be planned for future sort of care or services…For that individual child but for a population as well.” [FG12 P3]*
Benefits for reproductive planning and the broader family	*Quote 8: “…it would be good if they were aware of it for future children. Because some people have children very close together…And I’ve seen many cases where people end up with three children with the same genetic condition, because they had three children in four years or five years, before any symptoms were recognised.” [FG5 P4]* *Quote 9: “I think there are other implications for families in terms of your broader family, if you find things out that are inherited…” [FG1 P1]*

**Table 4 IJNS-10-00006-t004:** Potential challenges with gNBS.

Potential Challenges	Illustrative Quote(s)
Possible impact of gNBS on parents	*Quote 1: “When you’re handed a baby it’s so overwhelming, there’s so many new things already. I think adding another one then could just be overwhelming.” [FG4 P5]* *Quote 2: “…unless it’s completely accurate and we know what’s really likely to happen, it’s that, the alarm factor and who wants to be unnecessarily alarmed or feel unnecessarily alarmed if the accuracy of the initial testing is not as great as it should be?” [FG9 P1]* *Quote 3: “…but if the parent has a condition that can be pretty asymptomatic, what effect is that going to have on the parent if they find out they do have this condition even if they have had no complications of this condition prior and then they have a child with this condition and are very severely affected?...Would there be some like guilt there?” [FG2 P1]* *Quote 4: “…I’m just concerned that people who are presented with these situations very early on, is that going to affect their emotional attachment and bonding with their child?” [FG7 P2]*
Data security, privacy, discrimination and insurance concerns	*Quote 5: “…how’s the information going to be stored, like particularly in this digital world around cyber and security and all that stuff. How long will it be stored? Who’s got access to it and who could have access to it down the track?” [FG8 P3]* *Quote 6: “…if a child comes back saying that they have a higher chance of getting something, is that then going to affect their private health insurance? It’ll now either cost them more or they won’t be able to get covered for that?” [FG12 P5]*

**Table 5 IJNS-10-00006-t005:** When, who and what: Discussing and consent for gNBS.

	Illustrative Quote(s)
gNBS should be discussed during pregnancy	*Quote 1: “I feel when it comes time for the heel prick test, it’s just like, “Do you want to get the heel prick test done?” and you go, “Yeah, sure,” and your baby’s 48 h old, and you’ve just given birth and you’re a bit sleep deprived, and there’s not that considered decision making. So I think the earlier the information can be given, and time for consideration and research, the better.” [FG11 P3]* *Quote 2: “…maybe a public campaign about screening for particular conditions…not just people having babies, but educating the community about why this would be really important.” [FG13 P6]* *Quote 3: “…perhaps education packages for people who are trying to become pregnant as well, so they’ve got more of a chance to learn more about it…” [FG1 P4]* *Quote 4: “I kind of wonder whether that first appointment where women have just found out they’re pregnant or they’re planning to become pregnant, I wonder whether that would that be a good point where you are not necessarily, have any brochures yet. I mean obviously you get to the point in pregnancy where you are just getting so much information…” [FG1 P2]* *Quote 5: “I would prefer this sort of information to be disseminated and discussed probably in the third trimester of pregnancy…because it stays relevant. If it was maybe discussed any earlier it might sort of get lost in amongst other things that are happening during pregnancy…” [FG6 P3]* *Quote 6: “…the education just can’t depend on one person. It needs to be multiple people that you trust.” [FG13 P3]*
Consent for gNBS should be obtained during pregnancy	*Quote 7: “I don’t think it would be fair to put it onto someone who’s just given birth, whether we go ahead and test. So you need time. You need time to think and make a choice, an informed choice.” [FG14 P5]* *Quote 8: “I think you can maybe do an early consent and then re-confirm at collection because they may change their mind and decide they would want to do it when they said they didn’t want to do it or something.” [FG7 P1]*
Who should discuss gNBS	*Quote 9: “…in terms of who this information I suppose is given, or who gives this information, I would say it would be best coming from an obstetrician or maybe a midwife if that’s…who’s looking after the pregnancy. Maybe GPs…” [FG6 P3]* *Quote 10: “Maybe GPs, again depending on their knowledge and that sort of thing, they’re probably not the most appropriate person to be making this sort of I suppose recommendations. Maybe they can make the parents or the parent aware that this is available, but in terms of going into it further, they’re general practitioners, they’re not specialised.” [FG6 P3]* *Quote 11: “Midwife is a good option, but again, it would all depend on how much knowledge she has and how much she can share, and also the workload that she has. So that could be one of the drawbacks.” [FG13 P3]* *Quote 12: “…in an ideal world, it would be someone that had, like a genetic counsellor or someone with that type of background. But obviously that is probably more costly…” [FG1 P1]* *Quote 13: “I don’t think it really matters who does it, as long as it’s somebody that’s well educated and experienced in providing education to parents.” [FG7 P2]*
What parents need to know about gNBS	*Quote 14: “I think as much information as possible should be given to parents. It might be overwhelming but you would rather tell them than not tell them…” [FG6 P3]* *Quote 15: “…many people learn in many different ways, and that it won’t be a one size fits all…You have language barriers, cultural barriers, so it needs to be inclusive, you need to consult with community groups and present information in different ways if it’s going to be effective.” [FG3 P1]* *Quote 16: “…talking about what conditions are covered but what are the ones that possibly aren’t covered. There’s obviously thousands and thousands…you obviously can’t financially and logistically maybe test for all of those conditions. But there are, obviously, still letting people know there are conditions out there that we’re not able to test for at this point of time too… I think just letting them know it’s not a be-all, end-all, we’re not testing everything.” [FG4 P1]* *Quote 17: “…just some idea of the process that the testing will take place…if everything’s okay you won’t hear, or you will hear that everything’s okay. If anything comes up, you’ll be put in touch with a genetic counsellor, just some idea of what to expect in the next weeks after the testing.” [FG3 P1]* *Quote 18: “…how’s the information going to be stored…How long will it be stored? Who’s got access to it and who could have access to it down the track?” [FG8 P3]* *Quote 19: “I think written information is only as good as how capable the person that’s reading it is able to read, so yeah, a couple of different mediums would be good, and the ability to ask questions rather than just being, you know, just online, having a face-to-face conversation with someone and being able to ask questions that are pertinent to the parent.” [FG7 P2]*

**Table 6 IJNS-10-00006-t006:** Type of consent.

	Illustrative Quote(s)
Comparison with consent for other relevant healthcare interventions	*Quote 1: “…like vaccinations. They’re not really mandated, but they’re, I suppose people make it difficult not to have them. And then there’s the argument which is it can be negligent not to vaccinate your children. But the choice is still there.” [FG14 P2]* *Quote 2: “…in regards to consent I think it needs to be different because there is also, in and above the child or the baby themselves, there may be ramifications for the immediate and extended family of any results that are found and how they may be communicated to other members of the family.” [FG1 P4]* *Quote 3: “I don’t think the consent process should be any different because at the time you’re providing consent or not you would have gone through all the information about genomic testing and what it involves. So your, if you do have any questions, absolutely, feel free to ask but I think the consent process in itself shouldn’t really differ from the newborn screening test as it currently is.” [FG6 P3]*
Perceptions that gNBS should be mandated	*Quote 4: “…if it’s going to make such massive differences to children’s, baby’s lives, and potentially provide life-saving treatment, then do the parents really need to consent, or is it just something that should be done? But I guess superficially, yes, I think it should be a matter of consent.” [FG7 P2]* *Quote 5: “I believe an opt-out system is probably the best the best option I guess. There will be parents who have certain cultural or religious beliefs where they don’t want blood taken from their child. I’ve worked with people who will refuse certain medical procedures for their very young children because it’s their perception that it goes against their religion. And I think there is a risk if you don’t respect that, you start to get the fringe people out there that start making a big fuss on social media and in the news…you need to be seen to be doing the right thing…” [FG8 P5]* *Quote 6: “I think it would go without saying that you could always revoke consent at any stage even if the test has been sent off you can revoke it while it’s being processed so you don’t find out the results if that was what you desire.” [FG6 P2]*
Views about informed consent	*Quote 7: “I think the choice, it’s a good choice, both having the choice to do or not to do. I don’t think it should be mandated, if it’s introduced…because it’s a personal decision…I just think choice is probably a good thing.” [FG14 P2]*
Perspectives on whether both parents need to consent for gNBS	*Quote 8: “I’d say explicit consent and from both parents because of the implication that a genetic condition might have for either parent and also for siblings of that child…I actually believe that both parents should give consent in all areas. And so if you can’t obtain both parents’ consent at that time it should be either that the sample can be taken but not tested is maybe a possibility until consent can be obtained from the other parent.” [FG6 P4]* *Quote 9: “I would say if either parent gives consent maybe the testing could be done and it’s, and if the results could maybe be conveyed to the parent that did give consent…So if mum wants to know about it and dad doesn’t, for example, then maybe the testing could go ahead rather than not go ahead at all and only whoever gave consent in the first place is informed of those results.” [FG6 P3]*

## Data Availability

The datasets generated during the current study are available from the corresponding author upon reasonable request.

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
