# Peer review of "Australian Public Perspectives on Genomic Newborn Screening: Risks, Benefits, and Preferences for Implementation"

_2409-515X, 2024, doi:10.3390/ijns10010006_

Round 1

Reviewer 1 Report

Comments and Suggestions for Authors

The manuscript “Australian public perspectives on genomic newborn screening: Risks, benefits and preferences for implementation” provides an important evaluation of the public perception on genomic NBS, being important for those working in the field. The manuscript is well written, the structure is adequate and easy to read/understand. Nevertheless, a major revision is advised.

Major review

The type of information provided influences the perception of the public on genomics. Therefore, it is very important to the readers, so they can do an effective interpretation of the results provided on the manuscript, to know what was told on the video and on the interviews about gNBS. For example:

·         -Uncertainty of gNBS

·        - Sensitivity of gNBS

·         -Targeted gNBS for specific mutations/disorders or WES/WGS?

·        - What kind of mutations are going to be detected/reported? All variants? Only Pathogenic/likely pathogenic?

·        - Was presented to participants the possibility do detect VUS and its consequences?

·         ……

Minor reviews:

·         -Please consider to avoid referring names from the manuscript authors (in this case their initials) on the text. This explicit reference raises the question if a particular person that perform the task, night have influence on the result. It shouldn´t.

·        - Line 138: Please consider to refer that from the 155 only 75 answered.

Author Response

Major review

  1. The type of information provided influences the perception of the public on genomics. Therefore, it is very important to the readers, so they can do an effective interpretation of the results provided on the manuscript, to know what was told on the video and on the interviews about gNBS. For example:
  • Uncertainty of gNBS
  • Sensitivity of gNBS
  • Targeted gNBS for specific mutations/disorders or WES/WGS?
  • What kind of mutations are going to be detected/reported? All variants? Only Pathogenic/likely pathogenic?
  • Was presented to participants the possibility do detect VUS and its consequences?

Response: We have added the following sentences into the methods section:

“The video included information about what DNA and genomic sequencing are, and how genomic sequencing can lead to disease detection. It also outlined what stdNBS in Australia involves and briefly introduced the idea of using genomics in NBS.”

“The degree to which concepts such as the uncertainty and sensitivity of gNBS, targeted versus broader testing approaches, and the potential test outcomes were covered varied between focus groups depending on the discussion.”

We have also added to the limitations section, which now reads as follows:

“Certainly, one potential limitation of this study is that we deliberately provided limited information about how gNBS might be implemented prior to the focus groups so as not to bias their views. Topics such as the possibility of identifying variants of uncertain significance or late onset forms of disease using gNBS could be challenging for members of the public to understand.”

……

Minor reviews:

  1. Please consider to avoid referring names from the manuscript authors (in this case their initials) on the text. This explicit reference raises the question if a particular person that perform the task, night have influence on the result. It shouldn´t.

Response: We have elected to retain the references to names because specifying who performed the data collection and analysis are both points in the COREQ guidelines.

“Focus groups were facilitated by one researcher with support from a second and conducted via Zoom. Both researchers are skilled qualitative researchers with experience in focus group methodology and training in genetic counselling. None of the participants were known to the researchers.

Focus groups were recorded using the Zoom recording function and transcribed by the researcher who assisted with the focus groups. Interview transcripts were analysed by using inductive context analysis, whereby content categories are generated from the data, rather than predetermined [46]. Coding continued iteratively until all data relevant to the research question had been coded into categories and subcategories. Findings were discussed by both researchers involved in the focus groups to ensure rigour. Data analysis was managed using NVivo (released March 2023) [47].”

  1. Line 138: Please consider to refer that from the 155 only 75 answered.

We have moved the section which states how many participants were included to directly below this sentence so adding a clarification that 75 participated is no longer needed.

Reviewer 2 Report

Comments and Suggestions for Authors

The work presented is of scientific interest especially due to the complexity of the topic covered. There is a growing interest in gNBS linked above all to the reduction of diagnostic times and therefore of the diagnostic odyssey that parents face. However, this approach highlights many concerns linked above all to the impact that this investigation can have on the parents' experiences.

The method used is sufficiently correct and offers an in-depth view of the point of view of the subjects involved in the study.

It might be interesting to know the exclusion criteria that were considered in the recruitment of subjects.

One consideration concerns the fact that such complex topics to discuss would require a more in-depth educational aspect.

The implications relating to the possibility of identifying uncertain variants or late onset forms of disease using gNBS could perhaps be difficult to understand by non-healthcare subjects. These considerations could be reported in a section on the limitations of the study.

Finally, one of the strength of this work certainly concerns the need to implement information process for future parents, considering both the methodology and the most appropriate period for receiving such information.

Author Response

  1. It might be interesting to know the exclusion criteria that were considered in the recruitment of subjects.

Response: The only exclusion criteria were that they were less than 18 years of age or did not have the capacity to give consent for themselves to participate. We have added the following sentence at the start of the results: “No potential participants were excluded.”

  1. One consideration concerns the fact that such complex topics to discuss would require a more in-depth educational aspect. The implications relating to the possibility of identifying uncertain variants or late onset forms of disease using gNBS could perhaps be difficult to understand by non-healthcare subjects. These considerations could be reported in a section on the limitations of the study.

Response: We have added the following to the discussion:

“Certainly, one potential limitation of this study is that we deliberately provided limited information about how gNBS might be implemented prior to the focus groups so as not to bias their views. Topics such as the possibility of identifying variants of uncertain significance or late onset forms of disease using gNBS could be challenging for members of the public to understand.”

Round 2

Reviewer 1 Report

Comments and Suggestions for Authors I just checked the alterations made by the authors and in my opinion the manuscript is ready to be published.